# Disease- and Medication-Specific Differences of the Microbial Spectrum in Perianal Fistulizing Crohn’s Disease—Relevant Aspects for Antibiotic Therapy

**DOI:** 10.3390/biomedicines10112682

**Published:** 2022-10-23

**Authors:** Matthias Kelm, Simon Kusan, Güzin Surat, Friedrich Anger, Joachim Reibetanz, Christoph-Thomas Germer, Nicolas Schlegel, Sven Flemming

**Affiliations:** 1Department of General, Visceral, Transplant, Vascular and Pediatric Surgery, University Hospital of Wuerzburg, 97080 Wuerzburg, Germany; 2Unit for Infection Control and Antimicrobial Stewardship, University Hospital of Wuerzburg, 97080 Wuerzburg, Germany

**Keywords:** fistulizing Crohn’s Disease, microbial spectrum, anorectal abscess, perianal fistulas

## Abstract

Perianal fistulizing Crohn’s Disease (CD) with abscess formation represents an aggressive phenotype in Inflammatory Bowel Disease (IBD) with increased morbidity. Treatment is multidisciplinary and includes antibiotics, but knowledge about the microbial spectrum is rare often resulting in inadequate antimicrobial therapy. In this single center retrospective study, all patients who were operated due to perianal abscess formation were retrospectively analyzed and the microbial spectrum evaluated. Patients were divided into a CD and non-CD group with further subgroup analysis. 138 patients were finally included in the analysis with 62 patients suffering from CD. Relevant differences were detected for the microbial spectrum with anaerobic bacteria being significantly more often isolated from non-CD patients. In a subgroup-analysis of CD patients only, medical therapy had a relevant effect on the microbial spectrum since *Streptococcus groups* and *Enterobacterales* were significantly more often isolated in patients treated with steroids compared to those being treated by antibodies. In conclusion, the microbial spectrum of patients suffering from CD varies significantly from non-CD patients and immunosuppressive medication has a relevant effect on isolated pathogens. Based on that, adaption of antibiotic treatment might be discussed in the future.

## 1. Introduction

Crohn’s Disease (CD) represents a major socioeconomic burden with increasing incidences worldwide. However, despite that relevance, its pathophysiology is still not understood in detail and no curative therapy exist. Since CD is highly heterogenous regarding its phenotype, the disease can affect the whole intestine with local inflammation, strictures, or penetration. In case of penetrating or fistulizing CD, the main localization for abscess formations is either intraabdominal or perianal. While there exists robust data on the pathophysiology and treatment of intraabdominal abscesses including the microbial spectrum, little is known about perianal abscesses formations associated with CD despite high rates of incidence and prevalence. In general, perianal disease extent with fistulas and abscess formation implies a more aggressive phenotype with increased morbidity [1], while therapeutic strategies for perianal abscess formation depend on the disease extent and are usually multidisciplinary [2]. 20–43% of patients with CD develop perianal fistulas with many of those being usually complex with one third of patients having recurrent lesions [3,4]. While uncomplicated perianal fistulas can be treated systemically and/or by seton-drainage, abscess formation usually requires urgent surgery. Furthermore, due to the anti-inflammatory or immunosuppressive medication of many patients, additional antibiotic treatment is often included for optimal therapy. Despite a lack of robust evidence, current guidelines recommend the use of antibiotics for simple and complex fistulas in patients with CD [5]. Therefore, antibiotics still play a relevant role in perianal Crohn-associated fistulas and abscess formations, but while the microbial spectrum of perianal abscess formations remains to be largely unknown, no clear recommendation for the use of a specific antibiotic category exists [6]. Similarly, in non-CD abscesses antibiotic courses following drainage of the perianal abscess varied highly including the length of the treatment as demonstrated in a meta-analysis [7]. However, while the positive effect of additive antibiotic treatment on drained anorectal abscesses has been demonstrated and its use is recommended, adaptation of the antibiotic category on the microbial spectrum is highly important to avoid overtreatment and the creation antibiotic resistance. Furthermore, the antibiotic treatment should not be delayed and should consider individual patient characteristics, such as comorbidities and medication, as well as the potential spectrum of microbes. This is especially relevant for patients with CD who suffer from perianal fistulas and abscess formation since they are often under immunosuppressive or anti-inflammatory medication. However, current evidence on the microbial spectrum of anorectal abscess formations is low and empiric antibiotic treatment often insufficient [5]. While some studies detected microbes such as gram-negative bacilli and gram-positive cocci such as *E. coli* or *streptococci* for anorectal abscess formation in general, no data exist on the microbial spectrum of patients with CD who suffer from perianal fistulas and abscesses [8]. Therefore, the aim of this study was to detect the microbial spectrum of anorectal abscess formations from patients with CD in comparison to patients without CD and to evaluate the effect of anti-inflammatory or immunosuppressive medication on the spectrum of local microbes to improve current evidence and optimize antibiotic treatment regimens for this relevant complication.

## 2. Materials and Methods

### 2.1. Study Population

In this retrospective single-center study, all patients who received excision of a perianal abscess formation at the Department of General, Visceral, Transplant, Vascular and Pediatric Surgery of the University Hospital of Wuerzburg between 2012 and 2019 were included. Identification was done retrospectively based on coding data. In case of the detection of a fistula, a seton was placed according to current guidelines. All patients who received an intraoperative swab to detect the microbial spectrum were included and divided into two subgroups depending on the presence or lack of Crohn’s Disease, while all patients without an intraoperative swab were excluded, as well as patients under the age of 18. Diagnosis of anorectal abscess formation was usually made clinically and/or based on endosonography. Besides the microbial spectrum and the applicated antibiotic therapy, sociodemographic and clinicopathological data including the history of CD, immunosuppressive and/or anti-inflammatory medication as well as co-morbidities, such as COPD, diabetes and smoking were identified and collected for each patient from patient records. The distinction between CD and non-CD was made based on patient history as well as intraoperative rectoscopy to evaluate the presence of a proctitis. 

### 2.2. Microbial Spectrum

Intraoperative swabs were taken from each abscess formation for further microbiological workup. Microbial diagnostics included microbial cultures and were processed and analyzed by the Institute of Hygiene and Microbiology at the University of Wuerzburg in accordance with national and international guidelines and by CE certified tests in keeping with instructions of the manufacturers [9]. Following microbial workup, microbes were divided into six subgroups (*Streptococcus* spp., *Staphylococcus* spp., *Enterobacterales*, *Enterococcus* spp., and fungi, as well as others) and results were interpreted and analyzed for antibiotic resistances and usage according to epidemiological data routinely collected by the in-house antibiotic stewardship team of the University Hospital of Wuerzburg.

### 2.3. Anti-Inflammatory and Immunosuppressive Medication

To evaluate the effect of immunosuppressive or anti-inflammatory medication on anorectal abscess formation in patients suffering from CD, a subgroup analysis was performed. Patients were divided for preoperative medication (none, steroids, biologicals) and differences in the microbial spectrum were analyzed.

### 2.4. Statistical Analysis

Statistical analysis was performed using IBM SPSS 28.0 (IBM SPSS, Armonk, NY, USA). Descriptive data are presented as median with range or total numbers with percentage. Differences in patient characteristics were assessed by Chi-Square test, Fisher’s exact test, or ANOVA test in accordance with data scale and distribution. A *p*-value of <0.05 was considered statistically significant.

### 2.5. Ethical Approval

For this study, ethical approval was obtained from the Ethics Committee of the University of Wuerzburg, Germany (Approval Number: 2022070601).

## 3. Results

### 3.1. Patient Characteristics

In this retrospective single-center study, all patients who received surgery due to perianal abscess formation at the Department of General, Visceral, Transplant, Vascular and Pediatric Surgery at the University Hospital of Wuerzburg between 2012 and 2019 were evaluated (n = 532). Of those, 138 received a microbial swab intraoperatively and were included in the study with 62 patients suffering from CD while patients without intraoperative swab were excluded (Figure 1). As presented in Table 1, patients suffering from CD were significantly younger with a mean age of 25.4, compared to patients without CD (42.3, *p* < 0.001), and had a significantly lower BMI ((27.4 vs. 23.3, *p* < 0.001). In addition, levels of hemoglobin and c-reactive protein (CRP) were lower in the group of CD patients compared to the control group. Regarding co-morbidities, no differences were detected, and smoking habits were also comparable between both groups (30.3% vs. 40.3%, *p* = 0.45). Further analysis revealed no statistically significant differences between both groups regarding the ASA classification, as well as leukocytes levels.

### 3.2. Microbial Spectrum

To identify differences of the microbial spectrum in CD patients compared to non-CD patients, intraoperative swabs were analyzed according to national and international guidelines. Microbial analysis of intraoperative swabs revealed negative results in five patients per group (6.6% vs. 8.1%, *p* = 1.0). In the remaining patients, a total number of 67 pathogens could be identified with a median amount of two pathogens per patient. Regarding the isolated microbes, no differences between both groups were seen for streptococci, including the viridans group, as well as groups a, b, c, and g (23.7% vs. 32.3%, *p* = 0.448; 11.8% vs. 11.3%, *p* = 1.0). Similarly, rates of detected *Staphylococci* spp. were comparable between CD patients and non-CD patients (5.3% vs. 11.3%, *p* = 0.21), but the number of isolated anaerobic bacteria were statistically significantly increased in the control group compared to suffering from CD (46.1% vs. 21.0%, *p* = 0.002). While rates of *Enterococcus* spp. trended to be increased in patients with CD without reaching statistical significance (1.3% vs. 8.1%, *p* = 0.09), no differences were observed between both groups regarding fungal pathogens (1.3% vs. 4.8%, *p* = 0.33) (Table 2).

### 3.3. Effect of Anti-Inflammatory and Immunosuppressive Medication

To evaluate the effect of immunosuppressive and anti-inflammatory medication on perianal abscess formations including the microbial spectrum, a subgroup analysis was performed for all patients suffering from CD. Seven patients received other medication and were excluded from the analysis; thus, 55 patients were finally included with 17 patients receiving steroids and 27 patients antibody-based medication (biologicals) whereas eleven patients did not receive any medication (Table 3). Similarly to before, rates of detected streptococci from the viridans group (*p* = 0.82) as well as *Staphylococcus* spp. (*p* = 0.58) and anaerobic bacteria (*p* = 0.46) were comparable between all groups. However, patients with steroids showed a significantly increased expression of microbes from Streptococcus groups a, b, c, and g compared to the other two groups of patients without medication or antibody-based medication (29.4% vs. 0% and 7.4%, *p* = 0.037). In addition, infection with *Enterobacterales* tended to occur less often in patients with antibody-based medication compared to patients without medication or receiving steroids (33.3% vs. 54.6% and 64.7%, *p* = 0.11). No differences were observed for *Enterococcus* spp. and fungal pathogens.

### 3.4. Antibiotic Resistance

For the identification and susceptibility testing, VITEK 2 by bioMéreiux was used, and the Institute of Hygiene and Microbiology goes along with the breakpoints provided by EUCAST (European Commission on Antimicrobial Susceptibility Testing) [10]. *Enterobacterales* was the most frequent isolated pathogen in CD patients, therefore, we evaluated the initial empiric therapy and its effectiveness against *E. coli* in detail. *Enterobacterales* was isolated in 28 patients of whom 15 received antibiotic therapy chosen by the attending physician. While most patients received either ciprofloxacin or cefpodoxim in combination with metronidazole, the different antibiotic therapies were sufficient in every case except for one patient (Table 4). 

## 4. Discussion

Anorectal abscess formation is a common disease which is usually based on anal fistulas. However, development of perianal fistulas and abscess formation in patients with CD is characterized as more aggressive phenotype with increased morbidity usually requiring a multidisciplinary treatment approach [6,11]. While the treatment of choice in case of the occurrence of anorectal abscesses is usually surgery, international guidelines recommend additive application of antibiotics, despite the lack of robust evidence [5,12]. In addition, common microbial spectrums, especially in case of immunosuppressive medication, are not well defined; thus, choosing an adequate antibiotic is difficult, resulting in an insufficient therapy for many patients suffering from fistulizing perianal CD. To define and identify the microbial spectrum of those patients to enable a targeted antimicrobial therapy, we performed a retrospective analysis of our cohort with patients who were operated on due to perianal abscess formation. Based on our data, we demonstrate significant differences of the microbial spectrum for patients suffering from CD in comparison to the control group, as well as a relevant effect of anti-inflammatory and immunosuppressive medication on the detected microbes.

In our retrospective study, patients suffering from perianal fistulizing CD showed a significantly altered microbial spectrum in comparison to the control group. While rates of anaerobic bacteria were significantly lower in CD (46.1% vs. 21%, *p* = 0.002), in case of anorectal abscess, formation in the context of CD expression of *Staphylococcus* spp. and *Enterococcus* spp. tended to be increased (5.3% vs. 11.3%, *p* = 0.21; 1.3% vs. 8.1%, *p* = 0.09). On the other hand, microbes, such as anaerobic bacteria and *Enterobacterales,* including *E. coli,* were identified most often in the control group, which is in line with other studies [13]. However, further subgroup analysis for patients suffering from CD revealed significant differences depending on their medical treatment. Interestingly, rates of streptococcus groups a, b, c, and g were significantly increased in patients who received steroids (29.4% vs. 7.4%, *p* = 0.037) whereas infection with *Enterobacterales* was relevantly decreased in patients who received antibody-based medication compared to patients receiving steroids or no medical treatment (33.3% vs. 64.7%, *p* = 0.11), respectively (Table 3). Therefore, our data not only demonstrate relevant differences regarding the microbial spectrum for patients suffering from CD, but also a significant effect of anti-inflammatory and immunosuppressive medication on pathogens of anorectal abscess formation.

In general, data on the microbial spectrum of patients with anorectal abscess formation is rare and limited to small studies [14,15]. Liu et al. identified in a retrospective analysis that *E. coli* and *K. pneumoniae* are the predominant pathogens in non-diabetic and diabetic patients with perianal abscesses, respectively [13]. While the discussion about additive antibiotic treatment for perianal abscess formation remains controversial, a randomized-prospective trial from Turkey including 183 patients concluded that additional antibiotic treatment following surgical drainage of the abscess does not influence fistula formation [16]. Similarly, Nunoo-Mensah et al. identified an advantage of additional antibiotic treatment only for patients with relevant co-morbidities [17]. Therefore, current guidelines recommend additional antibiotic treatment in non-CD abscesses only for patients at risk (diabetes, immunosuppression, HIV) [12]. To further evaluate the potential of antibiotic therapy on perianal fistula formation, results from the ongoing randomized prospective OFF trial (Oral Antibiotics for Anal Abscess) are expected soon (Trial number: NCT03643198)**.** In regard of fistulizing CD, robust evidence exists on the microbial spectrum of intraabdominal abscesses [18,19], however, little is known about clinically relevant pathogens in perianal fistulizing CD [5]. Despite that fact, antibiotic treatment is recommended in perianal fistulizing CD, but the applied antibiotics vary relevantly throughout clinical routine, therefore no robust recommendation can be given on the specific antimicrobial type which should be used in those patients. Furthermore, a growing body of evidence demonstrates increasing rates of antibiotic resistance with insufficient empirical first-line therapy resulting in enhanced length of hospital stay as well as increased healthcare-related costs [20,21,22]. Based on that, knowledge about the microbial spectrum is highly relevant to provide adequate patient care for patients with perianal fistulizing CD. In line with that, our data demonstrate that steroids resulted in significantly enhanced rates of streptococcus groups in patients with CD. While this interesting effect is not completely understood to date, further studies are necessary to evaluate this observation in more detail. Despite this, almost all patients were covered by adequate antibiotics in this study. The significant increase of anaerobic bacteria in the CD group is well treated by metronidazole and the unexplained raise of streptococci in the subgroup analysis can be addressed with various appropriate antibiotic agents. The in-house AMS team has been advocating a less fluoroquinolone and third generation cephalosporin based antibiotic policy, including third generation oral antibiotics, such as cefpodoxim. Beta-lactam/beta-lactamase inhibitor combinations like Ampicillin/Sulbactam would be ideally fitting if the well-described resistance on *Enterobacterales* would not be an obstacle. Therefore, to avoid antibiotic resistance and to optimize patient care, antimicrobial stewardship programs represent a powerful tool for continuous evaluation and surveillance [23,24].

In line with the heterogenous data on the expression of microbes presented before, when we compare our data of perianal abscesses to the microbial spectrum of intraabdominal abscess formations following fistulizing CD, interesting observations can be made. While microbial swabs of intraabdominal abscesses demonstrated significantly decreased rates of *Enterobacterales* and anaerobes [17,18], in case of perianal abscess formations in CD patients only anaerobic bacteria were significantly decreased but no differences in Enterobacterales were detected. Furthermore, an important aspect remains to be the influence of immunosuppressive or anti-inflammatory medication on microbial profiles. Interestingly, while no clear effect of those medications on microbial spectra were observed for intraabdominal abscesses based on fistulizing CD [18], we did identify statistically significant differences in case of perianal abscess formations in CD patients in regard of the immunossuppressive medication, as the increased expression of stretococci in patients receiving steroid-based medication (Table 3). However, even if the observed changes for other microbes, such as anerobic bacteria and *Enterobacterales,* or in case of intraabdominal abscess formation, were not statistical significantly different, clear trends were observed for the influence of those medications on microbial spectra in the specific patient cohort of patients with fistulizing CD. With regard to the heterogenous results on the effect or anti-inflammatory and immunosuppressive medication on microbial expression on intraabdominal abscess formations, as well as the limited data for perianal abscesses in general, further studies with a greater number of patients are necessary to identify the disease’s relevant effects of immunosuppressive and anti-inflammatory medications on microbes in detail, and to evaluate the impact on antibiotic resistances. Additionally, dietary aspects and preoperative antibiotic intake of patients should be evaluated since both aspects can affect microbiota and, thus, the microbial spectrum of abscess formations. Nevertheless, the presented observations confirm the general relevance of an individual adjusted antibiotic therapy in CD patients. In addition, the potential effect of immunosuppressive medications on microbes in patients suffering from CD underlines again the importance of specific programs for a continuous optimal patient-centered evaluation and surveillance of antimicrobial therapies.

Our study has important limitations including its retrospective character and single-center design. Furthermore, since not every patient suffering from anorectal abscess formation received an intraoperative swab, the number of patients included in our study is limited, which narrows final conclusions. In addition, our analysis did not include susceptibility profiles on all identified pathogens for all groups, and the specific effect of immunossuppressive therapy on antibiotic resistance could not be analyzed due to low numbers of cases. However, the presented data represent the first study comparing the microbial spectrum of patients with anorectal abscess formation comparing non-CD to CD patients and recognizing individual differences based on systemic medication. 

## 5. Conclusions

Perianal fistulizing CD represents a more aggressive phenotype with increased morbidity, making multidisciplinary treatment, including antibiotics, challenging. Here, we demonstrate that the microbial spectrum of patients suffering from CD might vary significantly from non-CD patients and that immunosuppressive medication has a relevant effect on the pathogens detected. Based on that, differences in microbial spectrums, especially in CD patients on antibody-based immunossuppressants, should be respected in daily clinical practice when evaluating antibiotic therapy. In addition, further studies are necessarily analyzing disease and medication-specific differences, including susceptibility patterns of the predominantly identified pathogens, to draw final conclusions. To do so, antimicrobial stewardship has the potential to control the development of microbial resistance and to supervise antibiotic therapies.

## Figures and Tables

**Figure 1 biomedicines-10-02682-f001:**
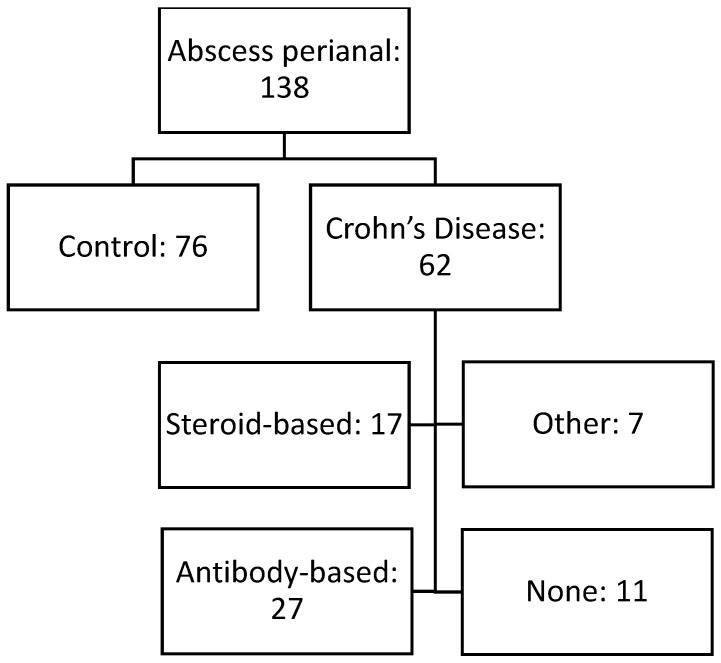
Study design.

**Table 1 biomedicines-10-02682-t001:** Patient characteristics. BMI body mass index, CVRF cardiovascular risk factors, COPD chronic obstructive pulmonary disease, ASA American Society of Anesthesiologists, Hb hemoglobin, CRP C-reactive protein.

	All (*n* = 138)	Control (*n* = 76)	Crohn’s Disease (*n* = 62)	*p*-Value
Age, years	31.3	42.3	25.4	<0.001
Sex (%) Female Male	34 (24.6) 104 (75.4)	18 (23.7) 58 (76.3)	16 (25.8) 46 (74.2)	0.84
BMI	24.8	27.4	23.3	<0.001
Co-morbidities (%) COPD Diabetes Smoking	15 (10.9) 5 (3.6) 48 (34.8)	8 (10.5) 3 (3.9) 23 (30.3)	7 (11.3) 2 (3.2) 25 (40.3)	1.0 1.0 0.45
ASA-Classification (%) > 2	116 (84.1)	57 (75)	59 (95.2)	
Infectious parameters, median Leukocytes CRP	11.5 3.7	12.4 3.8	10.7 3.3	0.34 0.014
Hemoglobin	13.4	13.8	13.1	0.018

**Table 2 biomedicines-10-02682-t002:** Microbial spectrum of perianal abscess formations of patients with Crohn’s Disease and without.

	All (*n* = 138)	Control (*n* = 76)	Crohn’s Disease (*n* = 62)	*p*-Value
Negative (*n*, %)	10 (7.2)	5 (6.6)	5 (8.1)	1.0
Skin microbiota (*n*, %)	39 (28.3)	28 (36.8)	11 (17.7)	0.014
Viridans group streptococci (*n*, %)	38 (27.5)	18 (23.7)	20 (32.3)	0.338
Streptococcus groups a, b, c and g (*n*, %)	16 (11.6)	9 (11.8)	7 (11.3)	1.0
*Staphylococcus* spp. (*n*, %)	11 (8.0)	4 (5.3)	7 (11.3)	0.21
Anaerobic bacteria (*n*, %)	48 (34.8)	35 (46.1)	13 (21.0)	0.002
*Enterobacterales* (*n*, %)	60 (43.5)	32 (42.1)	28 (45.2)	0.73
*Enterococcus* spp. (*n*, %)	6 (4.3)	1 (1.3)	5 (8.1)	0.09
Other (*n*, %)	12 (8.7)	4 (5.3)	8 (12.9)	0.14
Fungi (*n*, %)	4 (2.9)	1 (1.3)	3 (4.8)	0.33

**Table 3 biomedicines-10-02682-t003:** Microbial spectrum and influence of immunosuppressive therapy.

	None (*n* = 11)	Steroid-Based (*n* = 17)	Antibody-Based (*n* = 27)	*p*-Value
Negative (*n*, %)	0	2 (11.8)	3 (11.1)	0.51
Skin microbiota (*n*, %)	0	4 (23.5)	4 (14.8)	0.23
Viridans group streptococci (*n*, %)	4 (36.3)	5 (29.4)	7 (25.9)	0.82
Streptococcus groups a, b, c and g (*n*, %)	0	5 (29.4)	2 (7.4)	0.037
*Staphylococcus* spp. (*n*, %)	2 (18.2)	1 (5.88)	4 (14.8)	0.58
Anaerobic bacteria (*n*, %)	1 (9.1)	5 (29.4)	6 (22.2)	0.46
*Enterobacterales* (*n*, %)	6 (54.6)	11 (64.7)	9 (33.3)	0.11
*Enterococcus* spp. (*n*, %)	0	1 (5.9)	4 (14.8)	0.31
Other (*n*, %)	2 (18.2)	1 (5.9)	3 (11.1)	0.61
Fungi (*n*, %)	1 (9.1)	1 (5.9)	1 (3.7)	0.81

**Table 4 biomedicines-10-02682-t004:** Empiric antibiotic therapy and resistance profile for *E. coli* in patients with CD. 1 Ciprofloxacin plus metronidazole, 2 Cefpodoxim plus metronidazole, 3 Cefuroxim plus metronidazole, 4 Cefuroxim, 5 Cefotaxim, 6 Cefotaxim plus fluconazol.

	Antibiotic Therapy	Sensitive	Intermediate	Resistant	No Therapy
all CIP/MET ^1^ CEFPO/MET ^2^ CEFU/MET ^3^ CEFU ^4^ CEFOTAX ^5^ CEFOTAX/FLU ^6^	15 7 3 2 1 1 1	13 7 3 1 1 1	1 1	1 1	9

## Data Availability

The data presented in this study are available on request from the corresponding author. The data are not publicly available due to ethical reasons.

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
