# Peer review of "Disease- and Medication-Specific Differences of the Microbial Spectrum in Perianal Fistulizing Crohn’s Disease—Relevant Aspects for Antibiotic Therapy"

_biomedicines, 2022, doi:10.3390/biomedicines10112682_

Round 1

Reviewer 1 Report

The authors have performed a retrospective cohort study comparing microbe composition in patients who had a Crohn’s disease related abscess and non-CD related abscess. They found greater anaerobic bacteria in controls compared to patients with Crohn’s disease. The findings are novel given this research is limited. The clinical relevance of the findings is not entirely clear and limits the clinical relevance of the findings. There are a number of grammatical errors throughout the paper so a careful evaluation of this is required. I have the following additional comments:

- Page 3, study population – more details are required on how relevant patients were identified – was this done prospectively and kept in a database or retrospectively? If so, was this based on coding data? Please provide the total number of patients who had the drainage of the abscess so the number of total cases versus those who had a swab are known.

- Page 3, microbial specimen – was any standardised technique used to collect the specimen?

- Page 3, Results:

Study population - More details are needed on the nature of the abscesses in controls – were these all patients with a cryptoglandular or simple fistula? How was the distinction between Crohn’s disease and non-Crohn’s disease made – was it based on the complexity of the fistula tract or luminal disease? Is it possible that some patients were wrongly classified as non-Crohn’s if they had isolated perianal disease?

More details on inclusion and exclusion criteria for patients is needed

- Antibiotic resistance – was there any difference noted in resistance patterns between patients with Crohn’s disease and non-CD?

- Page 8, Conclusions:
The conclusions are too strong given that whilst there are differences noted in the microbes detected, it appears that current antibiotics therapy would cover this. Consider making considerable changes.

Minor points:

Page 1, line 35 – “oder” is not a word, consider changing.

Page 4, line 123 – “o” should be “no” please amend

Page 7, line 215 – “turkey” should be “Turkey”

Table 2 – skin germs is not medically appropriate – please revise

Table 4 – “Resistent” should be “Resistant”

Author Response

Dear Editor and Reviewers,

Thank you for the thorough review of our manuscript entitled Disease- and Medication-specific differences of the microbial spectrum in perianal fistulizing Crohn`s Disease – Relevant aspects for antibiotic therapy”. We thank you for many positive and constructive comments relating to our manuscript. In response to reviewers’ comments, we have made multiple text edits (highlighted in yellow in the revised paper) to address the reviewer`s concerns. We feel like these changes have strengthened the manuscript considerably and hope the revised manuscript is now acceptable for publication in Biomedicines.  In addition, a point-by-point response to reviewer comments is also provided.

If we can be of further assistance, please feel free to contact us directly. Thank you in advance for your consideration of our manuscript. We look forward to hearing from you and your office.

Sincerely,

Sven Flemming

Point-by-point response to the comments:

Editor Decision:

The fact that the study comes from one center is not necessarily a disadvantage. The aggregation of data from several centers with different levels of physician experience and potentially different patient management may, in my opinion, hurt the results. The retrospective nature of the study and the fact that not all patients with perianal abscess formation were included certainly constitute a disadvantage of the study. Nevertheless, the study is a valuable contribution to a subject in which an empirical approach prevails internationally. The authors could state in their conclusion part how much the results of the study influenced or will influence the management of their patients with perianal abscesses. What are their recommendations for dealing with patients in daily clinical practice?

Answer: We thank the Editor for the comment and appreciate the valuable feedback. We agree that our study has some limitations, however, we do believe that the results have great clinical relevance. Importantly, implementation of an Antimicrobial Stewardship helped us especially in complex cases of patients with immunosuppressants to find an individual based antibiotic therapy. In addition, the in-house AMS team has been advocating a less fluoroquinolone and 3rd generation cephalosporin based antibiotic policy, including 3rd generation oral antibiotics like cefpodoxim. We added this aspect to the manuscript (Line 228-236).

Furthermore and as demonstrated in our study, microbial species can vary significantly especially in patients with immunosuppressive therapy. While non-CD patients usually do not need antibiotics, guidelines recommend the use of antibiotics in CD patients suffering from anorectal abscess formations as stated in our introduction. However, microbial profiles seem to be similar between non-CD patients as well as CD patients without medication or steroid-based medication, therefore, at least for patients without immunosuppressive medication additive antibiotic therapy is not necessary based on our data. Nevertheless, especially CD-patients treated with antibody-based medication show a different microbial profile (more Staphylococcus and Enterococcus, less Enterobacterales and Streptococcus) which should be respected in clinical practice when decisions are made about antibiotic therapies. We edited our conclusion to underline this importance. (Line 23, 272-275)

Reviewer 1:

The authors have performed a retrospective cohort study comparing microbe composition in patients who had a Crohn’s disease related abscess and non-CD related abscess. They found greater anaerobic bacteria in controls compared to patients with Crohn’s disease. The findings are novel given this research is limited. The clinical relevance of the findings is not entirely clear and limits the clinical relevance of the findings. There are a number of grammatical errors throughout the paper so a careful evaluation of this is required. I have the following additional comments:

- Page 3, study population – more details are required on how relevant patients were identified – was this done prospectively and kept in a database or retrospectively? If so, was this based on coding data? Please provide the total number of patients who had the drainage of the abscess so the number of total cases versus those who had a swab are known.

Answer: We thank the Reviewer for this comment. All patients were identified retrospectively based on data coded by the surgeon. The total number of patients treated in our hospital due to perianal abscess formation between 2012 and 2019 was 532. We added those information to the manuscript (Line 71, Line 104).

- Page 3, microbial specimen – was any standardised technique used to collect the specimen?

Answer: Collection of microbial swabs was taken in a standardized way for each patient. Swabs were taken directly from the abscess without prior rinsing/lavage.

- Page 3, Results:

Study population - More details are needed on the nature of the abscesses in controls – were these all patients with a cryptoglandular or simple fistula? How was the distinction between Crohn’s disease and non-Crohn’s disease made – was it based on the complexity of the fistula tract or luminal disease? Is it possible that some patients were wrongly classified as non-Crohn’s if they had isolated perianal disease? More details on inclusion and exclusion criteria for patients is needed.

Answer: This is a relevant question and we thank the Reviewer for this comment. Not in every patient of the control group a fistula was detected but an anorectal abscess formation. As mentioned in the text (Line 72/73), only in case of the detection of a fistula a seton was placed. Furthermore, regarding the distinction of Crohn`s Disease and non-CD, the differentiation was done based on prior diagnosis/history of the patient and intraoperative examination. It is correct that some patients could be potentially classified as non-CD patients while suffering from CD, however, all patients received an additional rectoscopy to evaulate the presence of a proctitis. Thus, all patients of the control group (non-CD patients) did not show signs of a proctitis as well as had a negative history for CD, making it highly unlikely to be wrongly classified. We added those information to the manuscript. (Line 78/79).

- Antibiotic resistance – was there any difference noted in resistance patterns between patients with Crohn’s disease and non-CD?

Answer: We appreciate this comment since this is an interesting aspect. We did not analyze antibiotic resistance in non-CD patients but focused on CD patients only. In addition, no differentiation was done for CD patients with and without immunosuppressive therapy due to low numbers of patients. However, we do think that this aspect is relevant and should be definitely addressed in future studies. We added this limitation to the discussion (Line 263-265).

- Page 8, Conclusions:
The conclusions are too strong given that whilst there are differences noted in the microbes detected, it appears that current antibiotics therapy would cover this. Consider making considerable changes.

Answer: Based on this comment, we weakened our conclusion (Line 271). However, we do think that our results demonstrate relevant differences especially in case of immunosuppression. Also, while “overtherapy“ with antibiotics should be avoided, our study provides important information about microbial species in the special patient cohorts to enable a more targeted antibiotic therapy.

Minor points:

Page 1, line 35 – “oder” is not a word, consider changing.

Page 4, line 123 – “o” should be “no” please amend

Page 7, line 215 – “turkey” should be “Turkey”

Table 2 – skin germs is not medically appropriate – please revise

Table 4 – “Resistent” should be “Resistant”

Answer: We thank the Reviewer for these comments and corrected these grammatical mistakes in the manuscript.

Reviewer 2 Report

Kelm et al studied the perianal fistulizing Crohn’s disease, reporting the microbial spectrum of patients with and without CD and the influence of immunosuppressive medication and antimicrobial therapy. They suggested that antimicrobial therapy should be adapted to the microbial spectrum. 

Major comments:

In the Material & Methods section, the Ethical approval should be improved, describing the inclusion and exclusion criteria of patients (age range, co-morbidities, daily diet, common medication, or other diseases in past). The ethical approval number should be indicated. 

Table1:  Why the sex of patients is female? There is some explanation that excludes the male patients or the incidence of this pathology is in major in women?

Did you consider in your analysis the daily diet of patients? (vegetarian or not, e.g. Mediterranean diet). The diet is an important parameter to take into consideration in terms of gut microbiota, as well as the history in terms of antibiotics intake during infancy.

The legend of Table 2 should be improved. Does the information in this table correspond to non-medication?  Table 2 must be cited in the result section. 

What is mean “Non-relevant” in Table 2? What are the aspects that were excluded?

The results of Table 4 are not clear. Are these resistance results relative to E. coli or Staphylococcus? Did you take into consideration the role of antibiotics in gut microbiota? 

How do you evaluate antibiotic resistance? 

Author Response

Dear Editor and Reviewers,

Thank you for the thorough review of our manuscript entitled Disease- and Medication-specific differences of the microbial spectrum in perianal fistulizing Crohn`s Disease – Relevant aspects for antibiotic therapy”. We thank you for many positive and constructive comments relating to our manuscript. In response to reviewers’ comments, we have made multiple text edits (highlighted in yellow in the revised paper) to address the reviewer`s concerns. We feel like these changes have strengthened the manuscript considerably and hope the revised manuscript is now acceptable for publication in Biomedicines.  In addition, a point-by-point response to reviewer comments is also provided.

If we can be of further assistance, please feel free to contact us directly. Thank you in advance for your consideration of our manuscript. We look forward to hearing from you and your office.

Sincerely,

Sven Flemming

Point-by-point response to the comments:

Editor Decision:

The fact that the study comes from one center is not necessarily a disadvantage. The aggregation of data from several centers with different levels of physician experience and potentially different patient management may, in my opinion, hurt the results. The retrospective nature of the study and the fact that not all patients with perianal abscess formation were included certainly constitute a disadvantage of the study. Nevertheless, the study is a valuable contribution to a subject in which an empirical approach prevails internationally. The authors could state in their conclusion part how much the results of the study influenced or will influence the management of their patients with perianal abscesses. What are their recommendations for dealing with patients in daily clinical practice?

Answer: We thank the Editor for the comment and appreciate the valuable feedback. We agree that our study has some limitations, however, we do believe that the results have great clinical relevance. Importantly, implementation of an Antimicrobial Stewardship helped us especially in complex cases of patients with immunosuppressants to find an individual based antibiotic therapy. In addition, the in-house AMS team has been advocating a less fluoroquinolone and 3rd generation cephalosporin based antibiotic policy, including 3rd generation oral antibiotics like cefpodoxim. We added this aspect to the manuscript (Line 228-236).

Furthermore and as demonstrated in our study, microbial species can vary significantly especially in patients with immunosuppressive therapy. While non-CD patients usually do not need antibiotics, guidelines recommend the use of antibiotics in CD patients suffering from anorectal abscess formations as stated in our introduction. However, microbial profiles seem to be similar between non-CD patients as well as CD patients without medication or steroid-based medication, therefore, at least for patients without immunosuppressive medication additive antibiotic therapy is not necessary based on our data. Nevertheless, especially CD-patients treated with antibody-based medication show a different microbial profile (more Staphylococcus and Enterococcus, less Enterobacterales and Streptococcus) which should be respected in clinical practice when decisions are made about antibiotic therapies. We edited our conclusion to underline this importance. (Line 23, 272-275)

Reviewer 2:

Kelm et al studied the perianal fistulizing Crohn’s disease, reporting the microbial spectrum of patients with and without CD and the influence of immunosuppressive medication and antimicrobial therapy. They suggested that antimicrobial therapy should be adapted to the microbial spectrum. 

Major comments:

In the Material & Methods section, the Ethical approval should be improved, describing the inclusion and exclusion criteria of patients (age range, co-morbidities, daily diet, common medication, or other diseases in past). The ethical approval number should be indicated. 

Answer: We thank Reviewer 2 for this comment. As written in the manuscript, all patients who received excision of a perianal abscess formation and an intraoperative swab were included. Besides age >18, no exclusion criteria existed. We added this aspect to the manuscript (Line 74/75). Also, the ethical approval number was added.

Table1:  Why the sex of patients is female? There is some explanation that excludes the male patients or the incidence of this pathology is in major in women?

Answer: For greater clarity, only the proportion of female patients was listed. However, we now added also the number of male patients. (Table 1)

Did you consider in your analysis the daily diet of patients? (vegetarian or not, e.g. Mediterranean diet). The diet is an important parameter to take into consideration in terms of gut microbiota, as well as the history in terms of antibiotics intake during infancy.

Answer: This is an important aspect and we thank the Reviewer for this comment. We did not evaluate the diet of the included patients since no documentation of it was listed in patient records. However, since this is a valuable aspect we added it to our discussion. (Line 254-256).

The legend of Table 2 should be improved. Does the information in this table correspond to non-medication?  Table 2 must be cited in the result section. 

Answer: We thank the reviewer for this observation and corrected the legend as well as cited the table in the result section. (Line 124)

What is mean “Non-relevant” in Table 2? What are the aspects that were excluded?

Answer: We changed it to “Other“.

The results of Table 4 are not clear. Are these resistance results relative to E. coli or Staphylococcus? Did you take into consideration the role of antibiotics in gut microbiota? 

Answer: We thank the Reviewer for this observation. Table 4 represents resistance results relative to E. coli, the table legend was edited accordingly (Line 206). Regarding the role of antibiotics on gut microbiota, we did not evaluate patients for preoperative antibiotic intake since this is unusual for perianal abscess formation and no adequate documentation on that aspect existed in patient records. However, the role of microbiota on abscess formation in general is relevant and we agree with the Reviewer that influences on microbiota such as diet or antibiotics should be included in the analysis for future studies. We included this aspect in the manuscript (Line 253/254).

How do you evaluate antibiotic resistance? 

Answer: The institute of Hygiene and Microbiology of the university of Würzburg goes along with the breakpoints provided by the EUCAST (European Commission on Antimicrobial Susceptibility Testing), and for the identification and susceptibility testing VITEK 2 by bioMérieux is used. We added this aspect to the manuscript (Line 83/84, 169-171).

Round 2

Reviewer 1 Report

The authors have appropriately addressed all the suggested comments. Just two minor points:

Page 3, Microbial spectrum - please provide your description for how swabs were taken in this section

Page 3, line 79: "patiet" should be "patient"

Reviewer 2 Report

The authors answered all questions.